# Nanoconnectomic upper bound on the variability of synaptic plasticity

Thomas M Bartol Jr[1]*, Cailey Bromer[1], Justin Kinney[1,2†], Michael A Chirillo[3], Jennifer N Bourne[3‡], Kristen M Harris[3]*, Terrence J Sejnowski[1,4]*

[1]Howard Hughes Medical Institute, Salk Institute for Biological Studies, La Jolla, United States; [2]McGovern Institute for Brain Research, Massachusetts Institute of Technology, Cambridge, United States; [3]Center for Learning and Memory, Department of Neuroscience, The University of Texas at Austin, Austin, United States; [4]Division of Biological Sciences, University of California, San Diego, San Diego, United States

*For correspondence: bartol@ salk.edu (TMB); kmh2249@gmail. com (KMH); terry@salk.edu (TJS)

Present address: [†]Massachusetts Institute of Technology, Cambridge, United States; [‡]University of Colorado Denver, Denver, United States

Competing interests: The authors declare that no competing interests exist.

**Abstract** Information in a computer is quantified by the number of bits that can be stored and recovered. An important question about the brain is how much information can be stored at a synapse through synaptic plasticity, which depends on the history of probabilistic synaptic activity. The strong correlation between size and efficacy of a synapse allowed us to estimate the variability of synaptic plasticity. In an EM reconstruction of hippocampal neuropil we found single axons making two or more synaptic contacts onto the same dendrites, having shared histories of presynaptic and postsynaptic activity. The spine heads and neck diameters, but not neck lengths, of these pairs were nearly identical in size. We found that there is a minimum of 26 distinguishable synaptic strengths, corresponding to storing 4.7 bits of information at each synapse. Because of stochastic variability of synaptic activation the observed precision requires averaging activity over several minutes.

## Introduction

Synapses between neurons control the flow of information in the brain and their strengths are regulated by experience. Synapses in the hippocampus are involved in the formation of new declarative memories. Understanding how and why synaptic strengths undergo changes in the hippocampus is important for understanding how we remember facts about the world. A fundamental question is the degree of precision in the adjustment of synaptic strengths in view of the many sources of variability at synapses. In this study we provide an upper bound on the variability of synaptic plasticity and quantify a lower bound on the amount of information that can be stored at a single synapse.

Excitatory synapses on dendritic spines of hippocampal pyramidal neurons have a wide range of sizes. Anatomical measurements of the spine size, the area of the postsynaptic density (PSD), the number of AMPA receptors, the area of the presynaptic active zone and the number of docked vesicles in the presynaptic terminal are all highly correlated with each other and with physiological measurements of the release probability and the efficacy of the synapse (*Harris and Stevens, 1989*; *Lisman and Harris, 1994*; *Harris and Sultan, 1995*; *Schikorski and Stevens, 1997*; *Murthy et al., 2001*; *Branco et al., 2008*; *Bourne et al., 2013*). Thus, each of these individual characteristics is a correlate of synaptic strength. The sizes and strengths of these synapses can increase or decrease according to the history of relative timing of presynaptic inputs and postsynaptic spikes (*Bi and Poo, 1998*).

If experience regulates synaptic strength then one might expect that synapses having the same pre- and postsynaptic histories would be adjusted to have the same strength. But what would be the

**eLife digest** What is the memory capacity of a human brain? The storage capacity in a computer memory is measured in bits, each of which can have a value of 0 or 1. In the brain, information is stored in the form of synaptic strength, a measure of how strongly activity in one neuron influences another neuron to which it is connected. The number of different strengths can be measured in bits. The total storage capacity of the brain therefore depends on both the number of synapses and the number of distinguishable synaptic strengths.

Structurally, neurons consist of a cell body that influences other neurons through a cable-like axon. The cell body bears numerous short branches called dendrites, which are covered in tiny protrusions, or "spines". Most excitatory synapses are formed between the axon of one neuron and a dendritic spine on another. When two neurons on either side of a synapse are active simultaneously, that synapse becomes stronger, a form of memory. The dendritic spine also becomes larger to accommodate the extra molecular machinery needed to support a stronger synapse.

Some axons form two or more synapses with the same dendrite, but on different dendritic spines. These synapses should be the same strength because they will have experienced the same history of neural activity. Bartol et al. used a technique called serial section electron microscopy to create a 3D reconstruction of part of the brain that allowed the sizes of the dendritic spines these synapses form on to be compared. This revealed that the synaptic areas and volumes of the spine heads were nearly identical. This remarkable similarity can be used to estimate the number of bits of information that a single synapse can store, since the size of dendritic spines and their synapses can be used as proxies for synaptic strength.

Measurements in a small cube of brain tissue revealed 26 different dendritic spine sizes, each associated with a distinct synaptic strength. This number translates into a storage capacity of roughly 4.7 bits of information per synapse. This estimate is markedly higher than previous suggestions. It implies that the total memory capacity of the brain – with its many trillions of synapses – may have been underestimated by an order of magnitude. Additional measurements in the same and other brain regions are needed to confirm this possibility.

inherent variability, or conversely the precision, of this process? Due to the high failure rate and other sources of stochastic variability at synapses one might expect that the precision of changes in the strengths of these synapses *in vivo* to be low. The failure rate at synapses depends inversely on the strength, and therefore the size, of the synapse. On this basis the strengths of weaker, and therefore smaller and less reliable synapses, would be expected to be less precisely controlled than the larger and stronger synapses, which have a lower failure rate.

An ideal experiment to test for the precision of the changes in synaptic strength would be to stimulate *in vivo* the axonal inputs to two well-separated spines on the same dendrite to insure that they have the same presynaptic and postsynaptic history of stimulation. Nature has already done the experiment for us as pairs of spines on the same dendrite contacting the same axon satisfy this condition. Prior work suggests that such pairs of spines are more similar in size than those from the same axon on different dendrites (*Sorra and Harris, 1993*). Here we evaluated this axon-spine coupling in a complete nanoconnectomic three-dimensional reconstruction from serial electron microscopy (3DEM) (*Harris et al., 2015*) of hippocampal neuropil. We determined the similarity of synapses among pairs of spines and set an upper bound on the variability and the time window over which pre- and postsynaptic histories would need to be averaged to achieve the observed precision.

## Results

In a $6 \times 6 \times 5 \ \mu m^3$ complete 3DEM from the middle of stratum radiatum in hippocampal area CA1 (*Mishchenko et al., 2010*; *Kinney et al., 2013*; see Materials and Methods). We identified 449 synapses, 446 axons and 149 dendrites, which except for one identified branch point, are likely to originate from different neurons based on the size of the reconstructed volume and the obtuse branching angles of dendrites from these neurons (*Ishizuka et al., 1995*; *Megías et al., 1997*). We

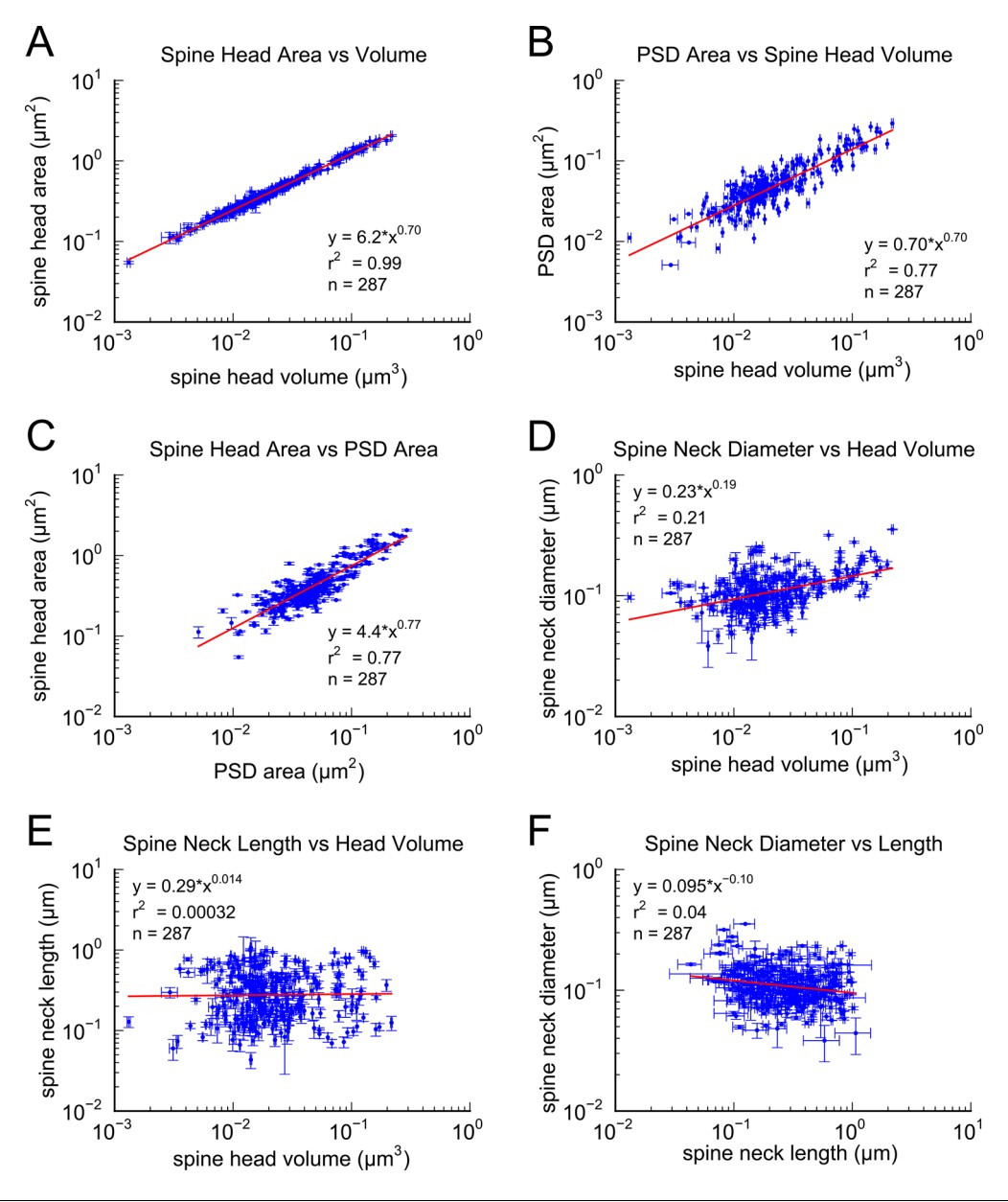

**Figure 1.** Correlations among metrics of dendritic spine morphology. Strong correlations were found between (**A**) Spine head area and spine head volume, (**B**) PSD area and spine head volume, and (**C**) Spine head area and PSD area. (**D**) Weak correlation was found between spine neck diameter and spine head volume. No correlation was found between (**E**) spine neck length and spine head volume and (**F**) spine neck diameter and spine neck length. Regression lines in red and error bars for each data point represent SEM based on multiple tracers who also edited each spine. Equations are based on the log-log distributions, with $r^2$ values indicated, and n=287 complete spines.

The following figure supplement is available for figure 1:

**Figure supplement 1.** Area of postsynaptic density plotted against spine head volume.

measured spine head volume and surface area, surface area of the postsynaptic density (PSD) adjacent to the presynaptic active zone, and spine neck volume, neck length and neck diameter at the 287 spines that were fully contained within the volume. We also quantified the number of vesicles at

the 236 spines and presynaptic boutons that were fully contained within the volume. The strong correlations between these metrics, the skewed shape of the frequency histograms, and the number of synapses per unit of volume (*Figures 1–3*), are consistent with previous observations (*Harris and Stevens, 1989*; *Schikorski and Stevens, 1997*; *Sorra et al., 2006*; *Bourne and Harris, 2011*; *Bourne et al., 2013*; *Bell et al., 2014*). To reduce error, we averaged over multiple independent

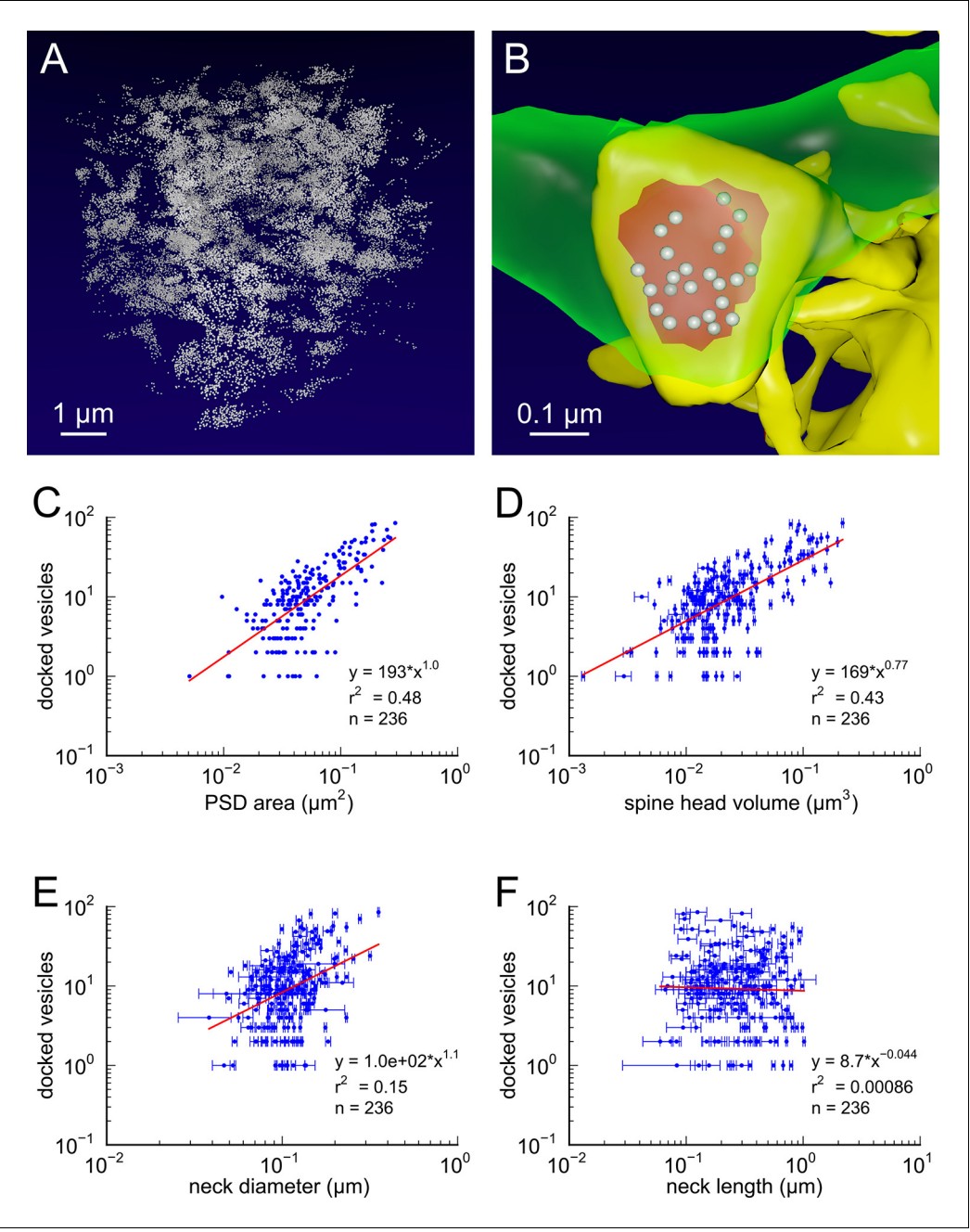

**Figure 2.** Presynaptic docked vesicle numbers are correlated with PSD areas, spine head volumes, and neck diameter, but not with neck length. (**A**) All 31,377 presynaptic vesicles. (**B**) *En face* view of the 24 docked vesicles (gray spheres) viewed through an axon (green) onto the PSD (red) of example spine (yellow). (**C**) Number of docked vesicles is correlated strongly with both PSD area and (**D**) spine head volume, weakly with (**E**) neck diameter, but is not correlated with (**F**) spine neck length. Regression lines, SEM (from multiple tracers), and $r^2$ are as in *Figure 1* n = 236 complete axonal boutons, each associated with one of the 287 complete spines. One human tracer marked PSDs and vesicles, hence no SEM for these two metrics.

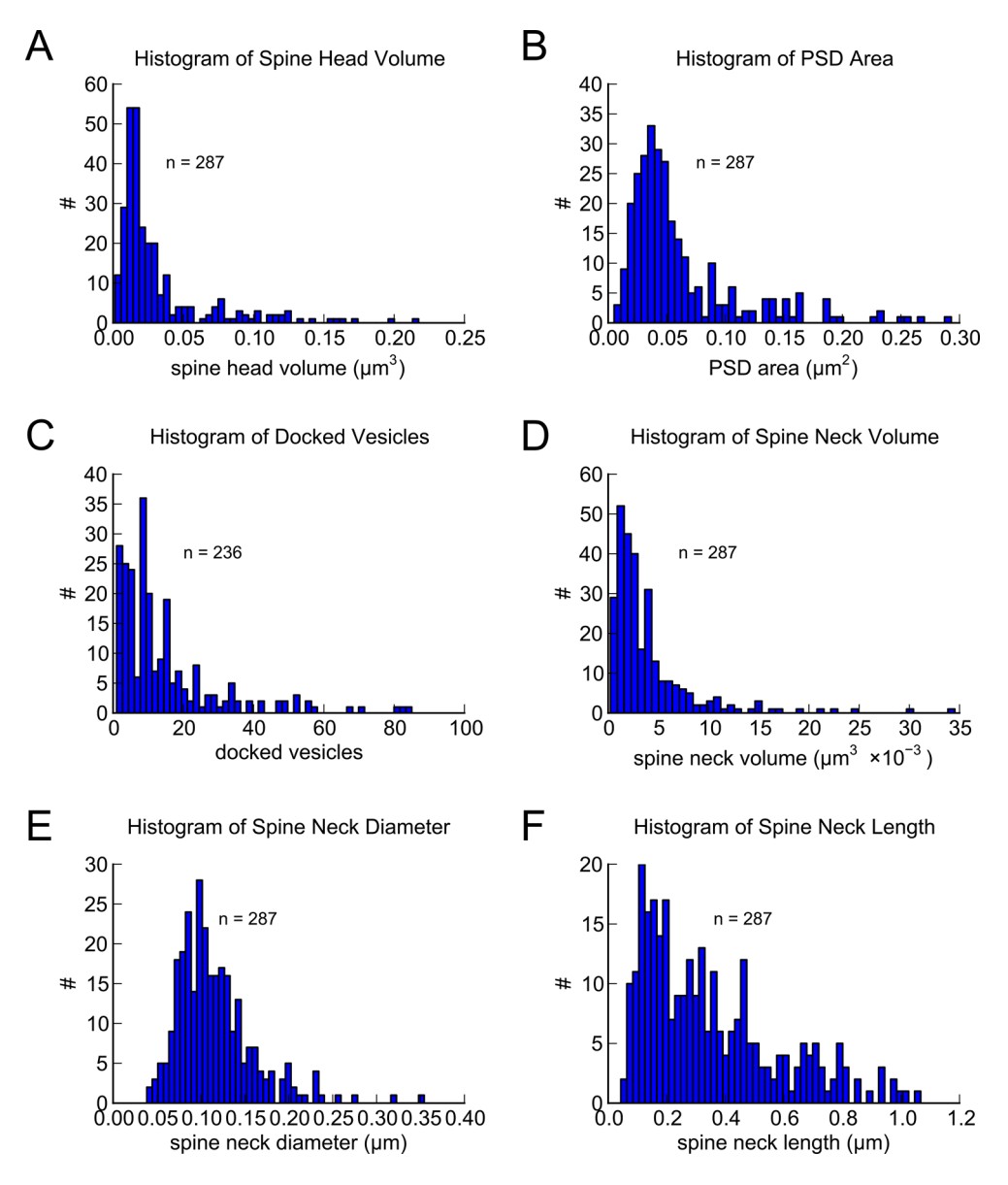

**Figure 3.** Morphometric analysis of 287 complete spines in reconstruction. Distributions of (**A**) spine head volumes, (**B**) PSD areas, (**C**) docked vesicles, (**D**) spine neck volumes, (**E**) spine neck diameters, and (**F**) spine neck lengths are highly skewed with a long tail.

The following figure supplement is available for figure 3:

**Figure supplement 1.** Spine measurement and estimation of measurement error.

spine volume measurements for each spine (*Figure 3—figure supplement 1*). We determined that the relationship between PSD area and spine head volume did not differ significantly across different dendrites (*Figure 1—figure supplement 1*). The correlation between spine head area and spine head volume accounted for 99% of the variance, despite the wide range in spine head shapes and dimensions (*Figure 1A*), which suggests that the accuracy of our measurements matched the precision of the spine. We also measured spine neck length, diameter, and volume and found a weak trend between the neck diameter (*Figure 1D*) and spine head volume but no correlation

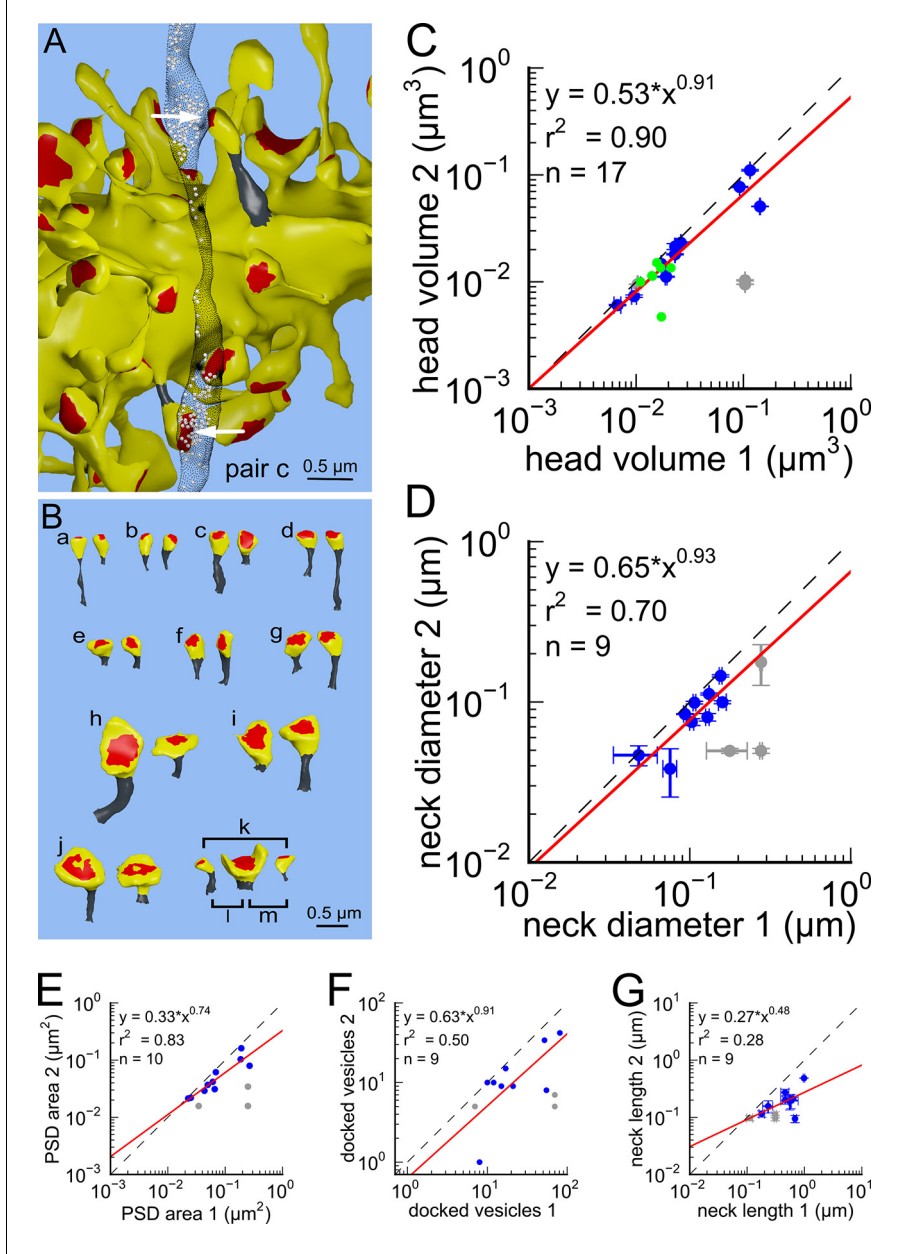

**Figure 4.** Spine head volumes, PSD areas and neck diameters, but not neck lengths, are highly correlated between pairs of axon-coupled same-dendrite spines. (**A**) Visualization of a pair of spines (gray necks) from the same dendrite (yellow) with synapses (red, indicated by white arrows) on the same axon (black stippling) with presynaptic vesicles (white spheres). (**B**) All axon-coupled same-dendrite spine pairs (colors as in **A**, pair c is elaborated in A). Strong correlations with slopes near 1 (dashed diagonal line) occur between paired (**C**) spine head volumes (slope = 0.91), (**D**) neck diameters (slope = 0.93), (**E**) PSD areas (slope = 0.74), and (**F**) docked vesicles (slope = 0.91); but not (**G**) spine neck lengths (slope = 0.48). Larger values from each pairing are plotted on the X axis. Regression lines (red) include the 10 a-j pairings (blue points) and 7 pairs from 2 additional animals (green points in (**C**)), but do not include triplet bouton pairings (k-m, gray points).

The following figure supplements are available for figure 4:

**Figure supplement 1.** Analysis of whole spine volume and spine neck volume of axon-coupled same dendrite spines.

**Figure supplement 2.** Analysis of spines paired randomly.

*Figure 4 continued on next page*

*Figure 4 continued*

**Figure supplement 3.** Axon-coupled same dendrite pairs a–f.

**Figure supplement 4.** Axon-coupled same-dendrite pairs g-m, illustrated in same way as in *Figure 4—figure supplement 3*.

between neck length (*Figure 1E*) and spine head volume, consistent with previous studies (*Harris and Stevens, 1989*; *Tønnesen et al., 2014*).

Next, we analyzed spine volumes according to their axonal connectivity and dendrite origin. Pairs of spines on the same dendrite that received input from the same axon ('axon-coupled'), were of the same size and had nearly identical head volumes (*Figure 4*). We compared this sample of 10 axon-coupled pairs on the same dendrite (*Figure 4B*, pairs a-j) to those identified on dendrites from the two additional animals (*Bourne et al., 2013*), for a total of 17 axon-coupled spine pairs. When plotted against one another, the paired head volumes were highly correlated with slope 0.91, and despite the small sample size, were highly significantly different from random pairings of spines (*Figure 4C* and *Figure 4—figure supplement 1A*, KS test p=0.0002). Similarly, there was a strong positive correlation between their paired neck diameters (*Figure 4D*), PSD areas (*Figure 4E*), and number of presynaptic docked vesicles (*Figure 4F*). These features of axon-coupled spines from the same dendrite spanned the distribution of the overall spine population (*Figure 3*). In contrast, the spine neck lengths (*Figure 4G*), and neck volumes (*Figure 4—figure supplement 1B*) of the pairs were not well-correlated indicating that regulation of neck length and neck volume are not important for synaptic strength.

The coupled triplet of synapses (*Figure 4C*, gray points 'k, l, m') are on three different spines along a single dendrite and receive synaptic input from a single multi-synaptic bouton. A larger central spine between two similar in size (*Figure 4B*, 'k, l, m') produces one same size pair ('k' ) and two different size pairs (' l', 'm'). This unusual configuration is probably driven by processes, such as competition for available resources, that differ from the other pairs (*Sorra and Harris, 1993*; *Sorra et al., 1998*). As one possibility, perhaps the the size of the larger postsynaptic spine was influenced by the larger size of the available pool of presynaptic vesicles in close proximity to its active zone. Excluding this triple synapse, the median value of the coefficient of variation of volume differences between pairs was CV = 0.083 and was as precise for small synapses as it was for large ones (*Figure 5*). This precision (i.e. low CV) suggests that accurately maintaining the size of every

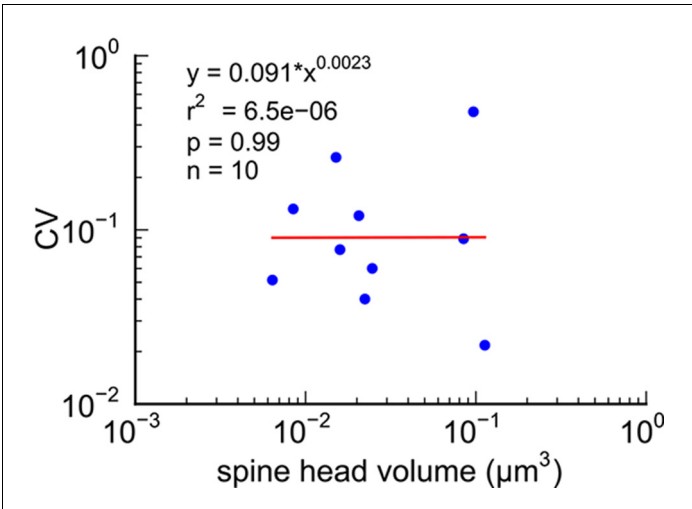

**Figure 5.** CV of axon-coupled spines on the same dendrite does not vary with spine size. There is no significant correlation, which implies that paired small synapses are as precisely matched as paired large synapses.

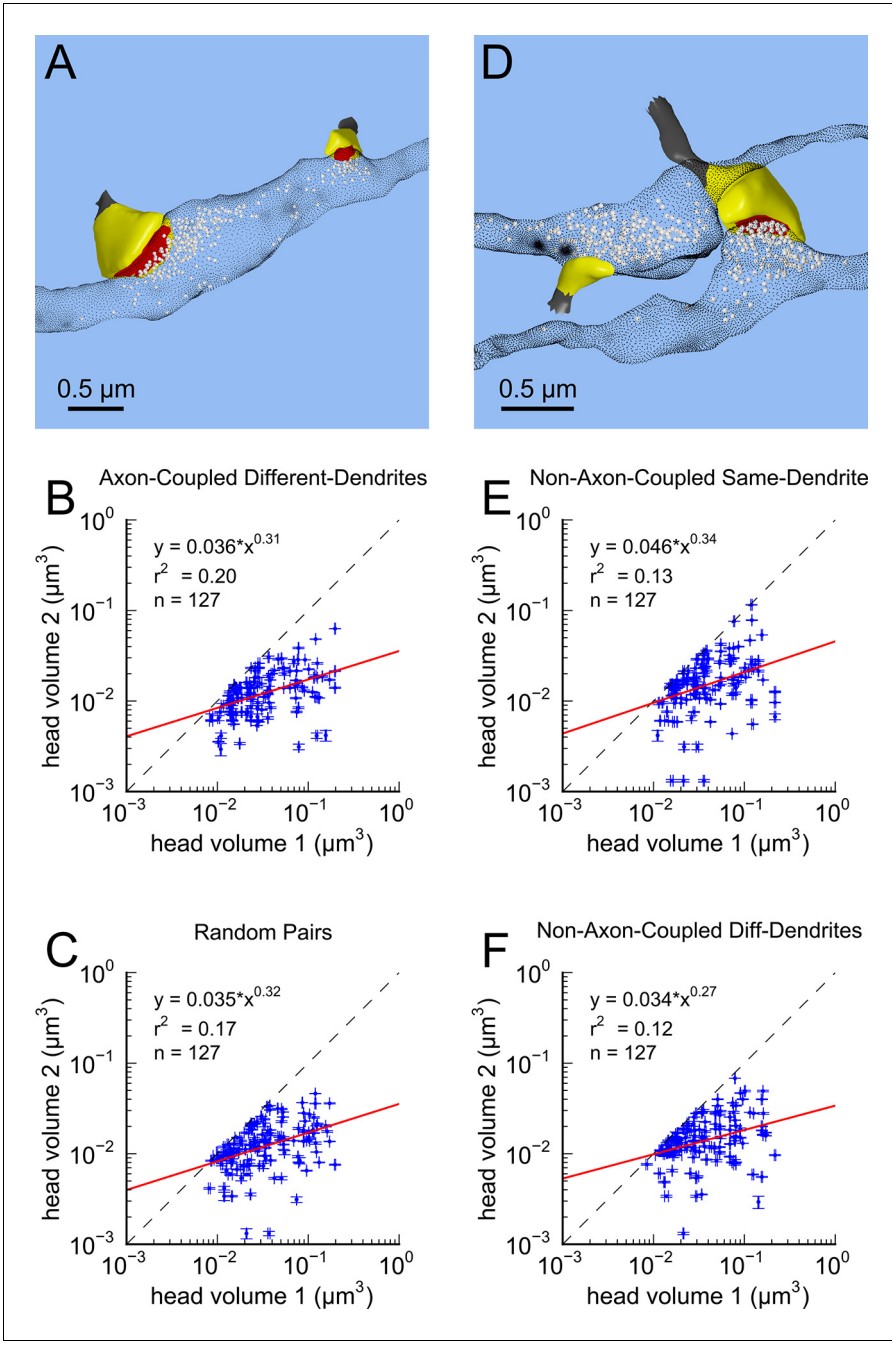

**Figure 6.** Paired spine head volumes are not correlated when they are not both axon and dendrite coupled. (**A**) Representative visualization and (**B**) plot showing lack of correlation between spine head volumes of all pairs of axon-coupled spines on different dendrites (n=127). (**C**) Similarly, randomly associated pairs of spine head volumes were not correlated. (**D**) Representative visualization and plots show lack of correlation between spine head volumes from randomly selected pairs (n=127) of non-axon-coupled spines (**E**) on the same or (**F**) different dendrites. Color scheme and regression analyses as in *Figure 4*.

The following figure supplements are available for figure 6:

**Figure supplement 1.** Morphologies of PSD, docked vesicles, and necks are not correlated when spines are not both axon and dendrite coupled.

**Figure supplement 2.** Difference in volume between pairs of axon-coupled spines exhibits a weak trend with separation distance.

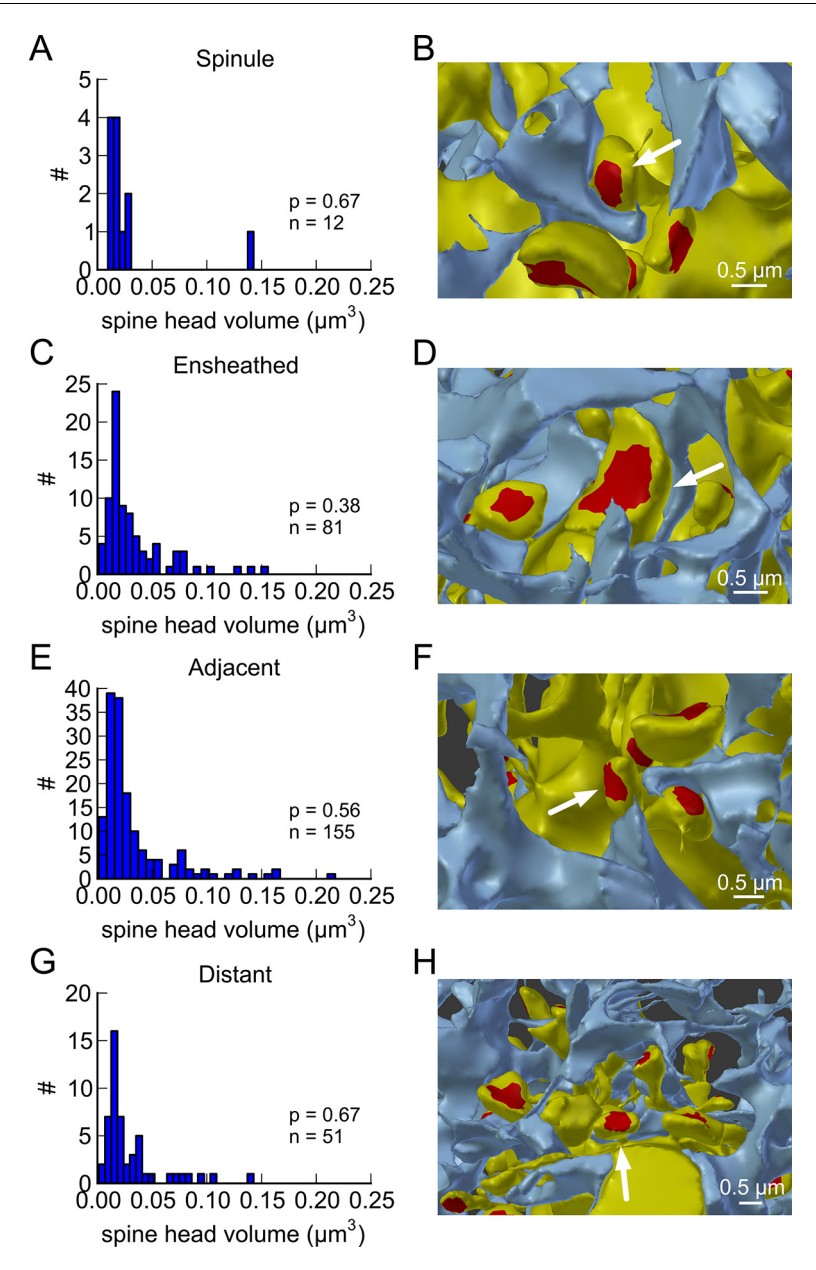

**Figure 7.** Proximity of the glial cell to axon-coupled dendritic spines on either the same or different dendrites. Proximity of astrocytic glial processes is not significantly correlated with spine head volumes of axon coupled pairs. (**A**) Histogram of spine head volume for spines that contain a spinule that is engulfed within the glial process (' spinule'). (**B**) Representation of an engulfed spinule. (**C**) Histogram of spine head volume for spines that are surrounded by and making contact with a glial process ('ensheathed'). (**D**) Representation of 'ensheathed' spine. (**E**) Histogram of spine head volume for spines that are proximal but not contacting a glial process ("' adjacent"'). (**F**) Representation of "'adjacent"' spine. (**G**) Histogram of spine head volume for spines that are distant from any glial process. (**H**) Representation of a spine "'distant"' from the glial process. The KS p value is shown on each inset and indicates that none of these distributions differ from the distribution for the whole population of spines.

synapse, regardless of size and strength, could be important for the function, flexibility and computational power of the hippocampus.

This near-identical size relationship does not hold for axon-coupled spines on different dendrites (*Figure 6B*, CV = 0.39, n = 127, example *Figure 6A*), nor for non-axon-coupled spines on the same

or different dendrites (*Figures 6E,F*, example *Figure 6D*) – all cases which would have had different activation histories. The volumes of axon-coupled different-dendrite spines are no different from the volumes of random pairs when plotted against one another (KS test p=0.94, *Figure 4—figure supplement 2A*, and *Figures 6B,C*) and the distribution of their sizes was no different from the whole population (KS test p=0.41). The number of docked vesicles for pairs on different dendrites (*Figure 6—figure supplement 1B*) is not different from random pairings (KS test p=0.08), nor are the neck diameters (*Figure 6—figure supplement 1C*, KS test p=0.06), nor the neck lengths (*Figure 6—figure supplement 1D*, KS test p=0.75). The size difference of pairs of axon-coupled spines on the same or different dendrites shows a weak trend with separation distance along the axon or dendrite (*Figure 6—figure supplement 2*). The sizes of pairs of axon-coupled spines on the same or different dendrites is unaffected by proximity of glia processes to the synapses (*Figure 7*) (*Ventura and Harris, 1999*; *Witcher et al., 2007*), or location of mitochondria in the axon (*Billups and Forsythe, 2002*).

We found that spine head volumes ranged in size over a factor of 60 from smallest to largest while the CV of any given size was 0.083 and was constant across the range of sizes. Measurements of these of 20 pairs allowed us to estimate the number of distinct spine sizes, and by extension synaptic strengths, that can be reliably distinguished across this range. Signal detection theory holds that at a Signal-to-Noise Ratio (SNR) of 1, a common detection threshold used in psychophysical experiments, an ideal observer can correctly detect whether a signal is higher or lower than some threshold 69% of the time (*Green and Swets, 1966*; *Schultz, 2007*). Put another way, if random samples are drawn from two Gaussian distributions whose areas overlap by 31%, an ideal observer will correctly assign a given sample to the correct distribution 69% of the time. Using this logic, we found that ~26 different mean synaptic strengths could span the entire range, assuming CV = 0.083 for each strength level, and a 69% discrimination threshold (*Figure 8*, see Materials and methods). These 26 distinct strength levels can be represented with 4.7 bits of information (i.e. $2^{4.7} \approx 26$) which means 4.7 bits of information that can be stored at each synapse as synaptic strength. At a

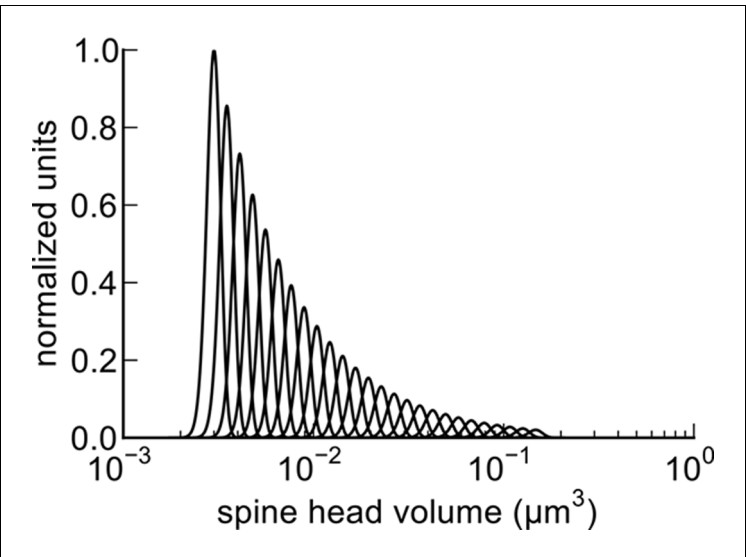

**Figure 8.** Distinguishable spine sizes. Over the factor of 60 range in spine head volumes from the data set there are 26 distinguishable intervals of spine sizes with a discrimination probability of 69% for each interval based on signal detection theory (*Green and Swets, 1966*; *Schultz, 2007*). The graph illustrates how distinct Gaussian distributions of spine sizes, each with a certain mean size and standard deviation, covers the entire range of spine head sizes on a log scale. The CV of each distribution is a constant value of 0.083 (*Figure 5*) and the intervals are spaced to achieve a total of 31% overlap with adjacent intervals giving a 69% discrimination threshold (see Materials and Methods). Note that the constant CV observed in the data set (*Figure 5*) means that the intervals appear uniform in width and spacing on a logarithmic scale. This is a form of non-uniform quantization which efficiently encodes the dynamic range of synaptic strengths at constant precision.

**Table 1.** Lower bounds on time window for averaging binomially distributed synaptic input to achieve CV = 0.083.

| Release probability ($p_r$) | Presynaptic spikes ($n$) | Averaging time (R = 1 Hz) | Averaging time (R = 25 Hz) |
|---|---|---|---|
| 0.1 | 1306 | 21.8 min | 52.2 sec |
| 0.2 | 581 | 9.68 min | 23.2 sec |
| 0.5 | 145 | 2.42 min | 5.8 sec |

discrimination threshold of 76% (corresponding to SNR = 2) there would be ~23 distinct strengths and 4.5 bits of information.

To explain the high precision observed in spine head volumes, we propose that time-window averaging smooths out fluctuations due to plasticity and other sources of variability including differences in the age of the synapses. To set a lower bound on averaging time, we chose to examine neurotransmitter release probability as a single source of variability. Let us first consider release caused by single action potentials, ignoring short-term plasticity. Release of this type can be analyzed using a binomial model in which n presynaptic action potentials, each with a probability $p_r$ of releasing one or more vesicles, leads to a mean number of releases $\mu = n*p_r$ having variance $\sigma^2 = n*p_r*(1-p_r)$. The coefficient of variation around the mean is CV = $\sigma/\mu$ = sqrt $[(1-p_r)/(n*p_r)]$ and can be compared with the measured values. Therefore, the number of spikes that are needed to reduce the variability to achieve a given CV is $n=(1-p_r)/(p_r*CV^2)$. *Table 1* gives averaging time windows $T = n/R$, where $R$ is spiking rate of the presynaptic axon, for representative values of $p_r$ and a range of spiking rates.

Accounting for other known sources of variability at dendritic spines would require even longer time windows. In particular, the impact of short-term plasticity during bursts of action potentials on the length of the time-window is complicated by the interplay of facilitation and depression. Synapses with a low initial $p_r$ (and corresponding long time-window in *Table 1*) exhibit marked facilitation and slowly depress during bursts (*Kandaswamy et al., 2010*; *Nadkarni et al., 2010*) which would shorten the time-window. But synapses with a high initial $p_r$ (and short time-window) only weakly facilitate, if at all, and quickly depress, which would lengthen the time-window.

## Discussion

Previous upper bounds on the variability of spine volume in the hippocampus, based on the whole spine volume (*Sorra and Harris, 1993*; *O'Connor et al., 2005*), underestimated the precision by including the spine neck volume (*Figure 4—figure supplement 1A*), which was not correlated between pairs of spines in our volume (*Figure 4—figure supplement 1B*). Our dense reconstruction included a complete inventory of every synapse in the reconstructed volume and in this respect was unbiased. Additional pairs of synapses from two other rats confirmed that this finding is not confined to a single brain. Of course, additional measurements in the hippocampus and other brain regions would be needed to confirm and extend this finding. The very high statistical significance of the finding (*Figure 4C*, KS test p=0.0002) despite a relatively small number of pairs in our sample implies a large effect magnitude, which would be much smaller if many more samples were needed to reach the same level of significance. To make this p value concrete, if 17 random pairs were chosen from all 287 synapses in the reconstructed volume, there is only a one in 5000 chance that the spine heads would be as precisely matched as the 17 axon-coupled pairs discovered here.

Previous studies have shown that there is a high correlation of the size of the spine head with the PSD area and numbers of docked vesicles (*Harris and Stevens, 1989*; *Lisman and Harris, 1994*; *Harris and Sultan, 1995*; *Schikorski and Stevens, 1997*; *Murthy et al., 2001*; *Branco et al., 2008*; *Bourne et al., 2013*). Since the correlations between the head sizes of axon-coupled pairs of spines is high, the high correlation between the PSD areas and numbers of docked vesicles observed in axon-coupled spines is not surprising (*Figures 4E and 4F*). However, it was unexpected to find that the spine neck diameters were also highly correlated between axon-coupled pairs of spines (*Figure 4D* $r^2$=0.70), since the correlation between spine head volumes and spine neck diameters is

not statistically significant (*Figure 1D*). Thus, there are at least two geometric aspects of the spine geometry that are under tight control of synaptic plasticity, which may reflect different aspects of synaptic function. The diameter of the spine neck may reflect the need for trafficking of materials between the spine shaft and spine head, which is known to be regulated by LTP and LTD (*Araki et al., 2015*).

Complementing our observations and analysis in the hippocampus, highly correlated $p_r$ at multiple contacts in the neocortex between the axon of a given layer 2/3 pyramidal neuron and the same target cell has been reported (*Koester and Johnston, 2005*). Our estimate of synaptic variability, based on spine head volume, is an order of magnitude lower. In a recent connectomic reconstruction of the mouse cortex, the similarity in the volumes of axon-coupled pairs of dendritic spines were statistically significant (*Kasthuri et al., 2015*). This observation is further evidence for the high precision of synaptic plasticity and suggests that the same may be true in other brain areas.

The axon-coupled pairs of synapses that we studied were within a few microns of each other on the same dendrite, which raises the question of how far apart the two synapses can be and still converge to the same size. Related to this question, two synapses from the same axon on two different dendrites of the same neuron might not share the same postsynaptic history. These questions cannot be answered with our current data due to the small dimensions and the fact that the position in the neuropil from which our reconstruction was taken makes it highly unlikely that any of the dendrites, other than the one branch point captured in the volume, belong to the same neuron (*Ishizuka et al., 1995*). Synaptic tagging and capture, in which inputs that are too weak to trigger LTP or LTD can be 'rescued' by a stronger input to neighboring synapses if it occurs within an hour (*Frey and Morris, 1997*; *O'Donnell and Sejnowski, 2014*), is much less effective when the synapses are on different branches (*Govindarajan et al., 2011*), which would tend to make two synapses from the same axon on different dendritic branches less similar. Probing these questions will require reconstructing a larger extent of hippocampus when a single axon can contact multiple dendritic branches of the same neuron (*Sorra and Harris, 1993*) or of other cells, such as layer 5 pyramidal cells, which can have 4–8 connections between pairs of neurons (*Markram et al., 1997*).

An unusual triple synapse from a single axon (*Figure 4B*, 'k, l, m') was excluded from the analysis because the presynaptic terminal was a single large varicosity filled with vesicles (i.e. an MSB) shared by three synapses, unlike the other pairs that had isolated presynaptic specializations (n=9), or an MSB shared by two synapses (n=8). It is possible that the large, central spine had an effectively larger pool of vesicles by virtue of proximity, whereas the two synapses on the outside had a more limited population to draw from, and the size of the postsynaptic spine was influenced by the size of the available pool. More examples are needed before we can reach any conclusions. Regardless of the explanation, our estimate of the variability would not be greatly affected by including these 3 additional pairs of synapses in the analysis.

How can the high precision in spine head volume be achieved despite the many sources of stochastic variability observed in synaptic responses? These include: 1) The low probability of release from the presynaptic axon in response to an action potential (*Murthy et al., 2001*); 2) Short-term plasticity of release of neurotransmitter (*Dobrunz et al., 1997*); 3) Stochastic fluctuations in the opening of postsynaptic NMDA receptors, with only a few of the 2–20 conducting at any time (*Nimchinsky, 2004*); 4) Location of release site relative to AMPA receptors (*Franks et al., 2003*; *Ashby et al., 2006*; *Kusters et al., 2013*) 5) Few voltage-dependent calcium channels (VDCCs) in spines that affect synaptic plasticity (smallest spines contain none) (*Mills et al., 1994*; *Magee and Johnston, 1995*); 6) VDCCs depress after back propagating action potentials (*Yasuda et al., 2003*); 7) Capacity for local ribosomal protein synthesis in some spines while others depend on transport of proteins from the dendrites (*Ostroff et al., 2002*; *Sutton and Schuman, 2006*; *Bourne et al., 2007*; *Bourne and Harris, 2011*); 8) Homeostatic mechanisms for synaptic scaling may vary (*Turrigiano, 2008*; *Bourne and Harris, 2011*); 9) Presence or absence of glia (*Ventura and Harris, 1999*; *Witcher et al., 2007*; *Clarke and Barres, 2013*); and 10) Frequency of axonal firing (*Callaway and Ross, 1995*).

One way that high precision can be achieved is through time averaging. Long-term changes in the structure of the synapse and the efficacy of synaptic transmission are triggered by the entry of calcium into the spine. A strategy for identifying the time-averaging mechanism is to follow the calcium. Phosphorylation of calcium/calmodulin-dependent protein kinase II (CaMKII), required for spike-timing dependent plasticity processes, integrates calcium signals over minutes to hours and is

a critical step in enzyme cascades leading to structural changes induced by long-term potentiation (LTP) and long-term depression (LTD) (*Kennedy et al., 2005*), including rearrangements of the cytoskeleton (*Kramár et al., 2012*). The time window over which CaMKII integrates calcium signals is within the range of time windows we predict would be needed to achieve the observed precision (*Table 1*). Similar time windows occur in synaptic tagging and capture, which also requires CaMKII (*Redondo and Morris, 2011*; *de Carvalho Myskiw et al., 2014*). These observations suggest that biochemical pathways within the postsynaptic spine have the long time scales required to record and maintain the history of activity patterns leading to structural changes in the size of the spine heads.

The information stored at a single synapse is encoded in the form of the synaptic strength, which reflects the pre- and postsynaptic history experienced by the synapse. But due to the many sources of variability, this information cannot be read out with a single input spike. This apparent limitation may have several advantages. First, the stochastic variability might reflect a sampling strategy designed for energetic efficiency since it is the physical substrate that must be stable for long-term memory retention, not the read out of individual spikes (*Laughlin and Sejnowski, 2003*). Second, some algorithms depend on stochastic sampling, such as the Markov Chain Monte Carlo algorithm that achieves estimates by sampling from probability distributions, and can be used for Bayesian inference (*Gamerman and Lopes, 2006*). Each synapse in essence samples from a probability distribution with a highly accurate mean, which collectively produces a sample from the joint probability distribution across all synapses. A final advantage derives from the problem of overfitting, which occurs when the number of parameters in a model is very large. This problem can be ameliorated by 'drop out', a procedure in which only a random fraction of the elements in the model are used on any given trial (*Wan et al., 2013*; *Srivastava et al., 2014*). Drop out regularizes the learning since a different network is being used on every learning trial, which reduces co-adaptation and overfitting.

We are just beginning to appreciate the level of precision with which synapses are regulated and the wide range of time scales that govern the structural organization of synapses. The upper bound on the variability that we have found may be limited by errors in the reconstruction and could be even lower if a more accurate method could be devised to compute the volume of a spine head, neck diameter, PSD area, number of docked vesicles, or other salient features of dendritic spines. Much can be learned about the computational resources of synapses by exploring axon-coupled synaptic pairs in other brain regions and in other species.

## Materials and methods

### Reconstruction of neuropil

Three separate 3DEM data sets were used in this study. Each of these data sets has been used for other purposes in prior studies. Images were obtained from serial thin sections in the middle of stratum radiatum of hippocampal area CA1 from three adult male rats (55–65 days old) (*Mishchenko et al., 2010*; *Bourne et al., 2013*). One set of images was used to make a dense model of 6 x 6 x 5 $\mu m^3$ of hippocampal neuropil and processed as previously described in a study of the extracellular space (*Kinney et al., 2013*). In this data set, we identified 13 axon-coupled synaptic pairs on 11 dendrites (*Figure 4—figure supplement 3* and *Figure 4—figure supplement 4*). The other two sets of images were part of a prior study (*Bourne et al., 2013*) in which subsets of dendrites and axons had been reconstructed. In this data set, we identified 7 axon-coupled synaptic pairs on 4 dendrites for a total of 20 axon-coupled dendrite-coupled spine pairs. To perform an accurate and robust geometric analysis of the dendrites, dendritic spines, axons, and glial processes, it was necessary to correct the reconstructed surface meshes for artifacts and make them into computational-quality meshes as described elsewhere (*Kinney et al., 2013*; *Edwards et al., 2014*).

The postsynaptic densities (PSDs) and presynaptic active zones (AZs) were identified in the serial section transmission electron microscopy (ssTEM) images by their electron density and presence of closely apposed presynaptic vesicles. We devised a method to segment the PSD-AZ features in the electron micrographs and mark their pre- and post-synaptic locations as subregions of the membrane in the final 3D mesh. To accomplish this, contours were hand-drawn on each serial section micrograph closely encompassing, as a single closed contour, the pre- and post-synaptic extent of the electron dense region. Taken together, the stack of contours for a given PSD-AZ forms a 3D

capsule which encloses the entire feature. VolRoverN (*Edwards et al., 2014*) was used to reconstruct the 3D surface of the capsule enclosing each PSD-AZ pair in 3D. Because these capsules enclose the intracellular domain of both the PSD and AZ they also overlap with the pre- and post-synaptic membrane associated with these subcellular features. Each of these closed capsules was then used as a '3D lasso' to tag mesh triangles of the pre- and post-synaptic membrane contained within the lasso, marking the enclosed membrane area as a synaptic contact region—PSD postsynaptically and AZ presynaptically. *Figure 3—figure supplement 1A* shows a postsynaptic contact area labeled in red on a dendritic spine.

The reconstructed neuropil models were then visualized and analyzed using Blender, a free, open-source tool for 3D computer graphics modeling (http://blender.org). A total of 449 synaptic contacts were found in the dense reconstructed volume of neuropil. We excluded a number of synapses from the analysis if they were partially clipped by the edge of the data set (142), or were shaft synapses (20) leaving 287 valid synapses on dendritic spines in the dense model. An additional 70 spines were excluded from the analysis of axon-coupled spines as the axon which contacted these spines did not contact any other spines within the reconstructed volume. Example visualizations of the spines and axons, generated using Blender, are shown in *Figure 2A,B*, *Figure 3—figure supplement 1A*, *Figure 4A,B*, *Figure 4—figure supplement 3*, *Figure 4—figure supplement 4*, *Figure 6A,D*, *Figure 7B,D,F,H*.

## Segmentation of dendritic spines

Blender's functionality is user-extensible via a Python interface for creating add-ons. We created a Python add-on for Blender that enabled the selection of the mesh triangles of the dendrite corresponding to the spine head and whole spine of each individual spine. Our add-on tagged each selected set of triangles with metadata for the spine name and geometric attributes of the head, whole spine, and neck as described below.

The selection of the spine head was made by hand based on a standardized procedure in which the junction between the head and neck was visually identified as half-way along the concave arc as the head narrows to form the neck (see *Figure 3—figure supplement 1A*). To select the whole spine, a similar visual judgment was made to locate the junction where the neck widens as it joins the dendritic shaft.

Once the appropriate area was selected, the tool was designed to automatically create the convex hull of the selected region. The closed mesh formed by the Boolean intersection of the convex hull and the dendrite was used to determine the measured volume of the spine head or whole spine. The volume of the neck was calculated by taking the difference between these two measurements.

Areas were computed from the selected regions for spine head and whole spine. Active zone and postsynaptic density areas were calculated using regions that had been determined during the hand-drawn reconstruction phase described above.

Distances between spine heads along the axon were calculated as the Euclidean distance between the centroids of the PSD/AZ regions. Distances between whole spines along the dendritic shaft were calculated as the Euclidean distance between the spine necks to shaft junctions. Glial classification, mitochondria classification and shape classification were performed by hand using set criteria.

## Estimation of measurement error of spine head volume

Some error in the measurement of spine head volume is expected to occur in the human judgment required to segment the dendritic spines into whole spine, head, and neck. To estimate this error, the valid spines in the dense model were segmented and measured a total of four times per spine (twice each by two people). The standard error of the mean in spine head volume decreases with volume and is less than 5% for the majority of spines with a median error of about 1% (*Figure 3—figure supplement 1*). The head volumes in the other two data sets were only measured once.

## Segmentation of synaptic vesicles and estimation of docked vesicles

Synaptic vesicles in the presynaptic terminals, totaling 31,377 in number, were identified along with their 3D locations within the dense reconstruction. Of the 449 presynaptic terminals, we excluded 193 terminals from the analysis due to truncation at the edge of the volume, and 20 terminals at

shaft synapses, leaving 236 valid terminals. A visualization of all the synaptic vesicles in the reconstruction is shown in *Figure 2A*.

Positive identification of docked vesicles in these ssTEM data sets is problematic due to the thickness of the sections and density of the staining. To estimate docked vesicles, we counted the number of vesicles whose centers were located within 100 nm of the presynaptic membrane across from the postsynaptic density of a given spine. Of the 31,377 vesicles, 3437 were labeled as docked according to this criterion which yielded estimates in good agreement with previous estimates (*Harris and Sultan, 1995*; *Schikorski and Stevens, 1997*; *Figures 2B–D*). An *en face* view of the docked vesicles at one synapse is shown in *Figure 2B*.

## Statistical analysis

All statistical analysis and plots were generated using Python 2.7 (http://python.org) with NumPy, SciPy, and Matplotlib. The distributions of spine head volume, spine head area, spine neck volume, PSD area, and AZ area were highly skewed with a long tail at larger values (*Figure 1*). Consequently, all regression analysis was performed using Pearson's linear regression on the data after applying a log-normal transformation ($r^2$ values shown in *Figures 1–6*).

The coefficient of variation (CV) of the population of spine pairings (*Figures 4* and *6*) was calculated as the median value of the CVs of each individual pair. The CV of each individual pair is simply the standard deviation of the volumes of the pair divided by the mean volume of the pair (*Figure 5*).

Population distributions were highly skewed making it necessary to make comparisons of distributions using non-parametric methods. We used the two-sample Kolmogorov-Smirnoff (KS) test to make these comparisons in *Figures 4 and 6*.

## Estimation of number of distinguishable spine sizes and bits of precision in spine size

To estimate the number of distinguishable spine sizes and corresponding bits of precision we calculated the number of distinct Gaussian distributions of spine sizes, each with a certain mean size and standard deviation that together would cover and span the entire range of spine head sizes seen in *Figure 4A*. *Figure 5* demonstrates that it is reasonable to assume that the CV of each these sub-distributions is a constant value of 0.083. From this CV, the spacing between the mean values of each sub-distribution can be chosen to achieve a total of 31% overlap with adjacent sub-distributions giving a 69% discrimination threshold. A 69% discrimination threshold is commonly used in the field of psychophysics and corresponds to a Signal-to-Noise Ratio (SNR) of 1 (*Green and Swets, 1966*; *Schultz, 2007*).

The 69% confidence interval, *z*, of a Gaussian distribution is given by:

$$z = sqrt(2) * erf^{-1}(0.69)$$

The spacing, *s*, of adjacent intervals of mean, $\mu$, is given by:

$$s = \mu * 2 * CV * z$$

The number, *N*, of such distributions that would span the factor of 60 range of spine sizes is:

$$N = \log(60)/\log(1 + 2 * CV * z)$$

$$N = 26.3$$

The number of bits of precision implied by *N* distinguishable distributions is given by:

$$bits = \log_2(N)$$

$$bits = 4.72$$

*Figure 8* shows that ~26 distinguishable distributions can cover the entire range of spine sizes, implying that there are ~4.7 bits of precision in the spine size.

All data and software tools described here are available at:
http://www.mcell.cnl.salk.edu/models/hippocampus-spine-analysis-2015-1

## Acknowledgements

We are grateful to Dr. Mary Kennedy, Dr. Charles Stevens, Dr. Cian O'Donnell, and Dr. Krishnan Padmanabhan, for discussions on many aspects of synaptic spines and CaMKII, Josef Spacek, and Dylan Yokoyama for data acquisition, and Libby Perry and Robert Smith for serial sectioning and image acquisition. This research was supported by NIH grants NS21184, MH095980, and NS074644 (Kristen Harris), NS44306, P41-GM103712, MH079076 and the Howard Hughes Medical Institute (T. Sejnowski).

## Additional information

### Funding

| Funder | Grant reference number | Author |
|---|---|---|
| National Institutes of Health | NS44306 | Thomas M Bartol<br>Justin Kinney<br>Terrence J Sejnowski |
| National Institutes of Health | GM103712 | Thomas M Bartol<br>Cailey Bromer<br>Terrence J Sejnowski |
| National Institutes of Health | MH079076 | Thomas M Bartol<br>Justin Kinney<br>Terrence J Sejnowski |
| Howard Hughes Medical Institute | | Thomas M Bartol<br>Justin Kinney<br>Terrence J Sejnowski |
| National Institutes of Health | NS074644 | Michael A Chirillo<br>Jennifer N Bourne<br>Kristen M Harris |
| National Institutes of Health | MH095980 | Michael A Chirillo<br>Jennifer N Bourne<br>Kristen M Harris |
| National Institutes of Health | NS21184 | Michael A Chirillo<br>Jennifer N Bourne<br>Kristen M Harris |

The funders had no role in study design, data collection and interpretation, or the decision to submit the work for publication.

### Author contributions

TMB, JK, KMH, Conception and design, Acquisition of data, Analysis and interpretation of data, Drafting or revising the article; CB, Acquisition of data, Analysis and interpretation of data, Drafting or revising the article; MAC, Acquisition of data; Analysis and interpretation of data; Drafting or revising the article; JNB, Acquisition of data; Analysis and interpretation of data; Drafting or revising the article; TJS, Conception and design, Analysis and interpretation of data, Drafting or revising the article

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
