## [Decision Letter]

Thank you for submitting your work entitled "Nanoconnectomic Upper Bound on the Variability of Synaptic Plasticity" for peer review at *eLife*. Your submission has been favorably evaluated by Eve Marder (Senior editor) and three reviewers, one of whom, Sacha Nelson, is a member of our Board of Reviewing Editors. Another reviewer, Winfried Denk, also agreed to reveal his identity.

The reviewers have discussed the reviews with one another and the Reviewing editor has drafted this decision to help you prepare a revised submission.

Summary:

This elegant study makes a simple but important point. By completely reconstructing a small volume of hippocampus, the authors demonstrate that pairs of synapses between the same dendrite and axon have highly correlated sizes (and therefore from earlier work) strength. The finding, that synapses that share an axon and a dendritic branch are very similar in size, is a very important confirmation of the notion that it is the history of correlated pre- and post-synaptic activity that determines synapse size and strength. The data were carefully and creatively analyzed. The results are convincing because they have been reproduced in three animals (a very time consuming task) and because of the high p values.

Essential revisions:

1) The authors should revise the text to clarify how the analysis was carried out, and what can and what cannot be determined from the datasets studied.

*Reviewer 1*: "It would be surprising if synapses made by the same axon on different dendrites of the same neuron (especially if located at similar electrotonic distances) were not better correlated than synapses made onto different neurons. Perhaps this did not occur in the volume reconstructed or the volume was not large enough to assign dendrites to the same neuron. If so it would be useful to state this."

*Reviewer 2*: "The third paragraph of the Results reads like an orphan yet provides the estimate for how much information can be stored at hippocampal synapses. The rational and analysis should be described in the Results section in more detail which will make the data and the argument more understandable to a wider range of readers. Also a short discussion of how varying the discrimination threshold would affect the estimate would be useful."

*Reviewer 3*: "I am not sure I quite follow the logic of the estimate for the time or the number of action potentials the paired synapses would need to average over to achieve the degree of similarity seen. During pre-synaptic bursts the release probability could go quite high, removing the noise due to the stochastic release seen during single APs.

I found myself confused by the use of "dendritic branch" and "dendrite". If might be better to use "dendritic branch" exclusively when talking about the data in the paper since the small size of the data set does not allow the determination of whether two branches belong to the same cell or not.

I am also somewhat puzzled by the statement "Difference in volume between pairs of spines is not correlated with separation distance", which heads the caption of Figure 14. First of all, not observing a correlation does not imply that there isn't any. At best an upper bound could be given for the actual correlation, but for small numbers of observation that estimate would be rather soft. Furthermore, to me, it looks like there is a trend towards greater variability as the distance increases, which would indicate the involvement of a diffusible signal in the size determination.

Ordering the spines by size before correlation analysis is inappropriate since it induces a correlation between them in the sense that knowing something about spine 1 tells me something about spine 2. Please randomize and redo the analysis.”

2) The authors should strive to make the presentation more concise.

*Reviewer 1*: "My only major suggestion for improving the paper is to make it more concise. Many of the figures are largely redundant (two are not cited in the text)."

*Reviewer 3*: "The paper could benefit from pruning some of the figures."

Please note, that if you feel strongly that you want to keep some of the images that the reviewers questioned, they should be included as Supplements to the Figures they "duplicate".

---

## [Author Response]

*Essential revisions: 1) The authors should revise the text to clarify how the analysis was carried out, and what can and what cannot be determined from the datasets studied.* Reviewer 1: *"It would be surprising if synapses made by the same axon on different dendrites of the same neuron (especially if located at similar electrotonic distances) were not better correlated than synapses made onto different neurons. Perhaps this did not occur in the volume reconstructed or the volume was not large enough to assign dendrites to the same neuron. If so it would be useful to state this."*

We agree with this hypothesis. Due to the small extent of the volume it is highly unlikely that any of the dendritic segments belonged to the same neuron, making it impossible to test this idea with the present data. We have added this point to the Discussion.

Reviewer 2: *"The third paragraph of the Results reads like an orphan yet provides the estimate for how much information can be stored at hippocampal synapses. The rational and analysis should be described in the Results section in more detail which will make the data and the argument more understandable to a wider range of readers.*

Also a short discussion of how varying the discrimination threshold would affect the estimate would be useful."

We agree and have expanded this section in the third paragraph of the Results section.

Reviewer 3: *"I am not sure I quite follow the logic of the estimate for the time or the number of action potentials the paired synapses would need to average over to achieve the degree of similarity seen. During pre-synaptic bursts the release probability could go quite high, removing the noise due to the stochastic release seen during single APs.”*

We have now clarified the logic of this analysis in the Results section according to the reviewer’s suggestion. Our estimate is only meant to be an approximate upper limit on the number of spikes. The reviewer makes an important point about bursts of APs. Our analysis does not account for shortterm facilitation during bursts, which presumably would have similar affects on both synapses in a coupled pair.

“*I found myself confused by the use of "dendritic branch" and "dendrite". If might be better to use "dendritic branch" exclusively when talking about the data in the paper since the small size of the data set does not allow the determination of whether two branches belong to the same cell or not.”*

We agree it is confusing to switch between these terms, as they are actually synonyms. We believe readability is improved by using the term “dendrite” exclusively (rather than dendritic branch). We have also added text and references at the beginning of the Results section to justify our reasoning that all but one of the dendrites in our sample likely belong to different neurons.

“I am also somewhat puzzled by the statement "Difference in volume between pairs of spines is not correlated with separation distance", which heads the caption of Figure 14. First of all, not observing a correlation does not imply that there isn't any. At best an upper bound could be given for the actual correlation, but for small numbers of observation that estimate would be rather soft. Furthermore, to me, it looks like there is a trend towards greater variability as the distance increases, which would indicate the involvement of a diffusible signal in the size determination.”

We agree with this assessment and have reworded the Figure caption and discussion in the text.

“Ordering the spines by size before correlation analysis is inappropriate since it induces a correlation between them in the sense that knowing something about spine 1 tells me something about spine 2. Please randomize and redo the analysis.”

Note that we’re not ordering by size, we’re ordering by separation distance, so there is no additional information that would affect the correlation. We’re not sure if this is what the reviewer is saying here.

*2) The authors should strive to make the presentation more concise.* Reviewer 1: *"My only major suggestion for improving the paper is to make it more concise. Many of the figures are largely redundant (two are not cited in the text)."*

Reviewer 3: *"The paper could benefit from pruning some of the figures."*

We have reorganized the Figures as described:

Figure 5 is now Figure 1—figure supplement 1

Figure 4 is now Figure 3—figure supplement 1

Figure 9 has been reorganized and moved to Figure 4—figure supplement 1.

Figures 10, 7 and 8 are now Figure 4—figure supplement 2, Figure 4—figure supplement 3, and 4 respectively.

Figures 13, 14 are now Figure 6—figure supplement 1 and Figure 6—figure supplement 2, respectively.

This has reduced the number of primary figures from 16 to 8.